# The Association between Air Pollution and Sleep Duration: A Cohort Study of Freshmen at a University in Beijing, China

**DOI:** 10.3390/ijerph16183362

**Published:** 2019-09-11

**Authors:** Hongjun Yu, Panpan Chen, Shelby Paige Gordon, Miao Yu, Yangyang Wang

**Affiliations:** 1Department of Physical Education, Tsinghua University, Tsinghua Yuan Str., Beijing 100084, China; cpp16@mails.tsinghua.edu.cn; 2Mailman School of Public Health, Columbia University, New York, NY 10032, USA; spgordo2@gmail.com; 3Renmin University of China Libraries, Beijing 100872, China; 4Department of Sociology, Tsinghua University, Beijing 100084, China

**Keywords:** air pollution, AQI, fine particulate matter, sleep duration, youth

## Abstract

*Background:* Rising levels of air pollution in Beijing, China have become a serious environmental issue affecting human health, and young adults are experiencing high rates of insufficient sleep duration or a lack of sleep. Gaps in previous research remain regarding the relationship between air pollution and sleep duration among young adults. The purpose of this study is to assess the associations between air pollution and sleep duration among college students living in Beijing, China. *Methods:* We conducted follow-up health surveys on 16,889 freshman students enrolled at Tsinghua University over a five-year study period (2013–2018). Sleep duration was measured using the Chinese version of the Pittsburgh Sleep Quality Index (CPSQI), which has been validated in China to measure sleep duration. Corresponding levels of the average hourly air quality index (AQI), PM_2.5_ (µg/m^3^), PM_10_ (µg/m^3^), and NO_2_ (µg/m^3^) were gathered from data provided by the Beijing Municipal Ecological Environment Bureau in a closed site at Tsinghua University. Multilevel mixed-effects linear regression models were used to analyze the data. *Results:* One standard deviation increase in air pollution concentration in AQI, PM_2.5_, PM_10_, and NO_2_ was associated with a reduction in daily hours of sleep by 0.68 (95% confidence interval (CI) = 0.63, 0.73), 0.55 (95% CI = 0.51, 0.59), 0.70 (95% CI = 0.64, 0.76), and 0.51 (95% CI = 0.47, 0.54), respectively. *Conclusions:* Air pollution was associated with a reduction in sleep duration among freshman students living in Beijing, China. Replication of this study is warranted among various populations within China.

## 1. Background

The rise in fossil fuel use, industrialization, and population growth have caused air pollution to become a severe threat to global health [1]. The World Health Organization (WHO) Global Ambient Air Quality Database estimates that in 2018 ambient air pollution contributed to 7 million deaths worldwide [2]. Approximately 91% of the world’s population live in environments with fine particulate matter (PM_2.5_) air pollution levels exceeding the WHO guideline of 10 µg/m^3^ [2]. In recent decades, China has experienced significant economic reforms, rapid industrialization, and drastic urban development [3]. As a result, exposure to air pollution has become a dominant threat to public health in China. Between 2011 and 2014, the average level of PM_2.5_ in the North China Plain exceeded 600 µg/m^3^, the highest recorded in the region’s history [4]. In 2010, air pollution in China was the fourth leading cause of mortality and led to 1.2 million premature deaths, almost 40% of the global total [5]. Previous studies have shown that short- and long-term exposure to PM_2.5_ is associated with adverse health effects (i.e., cardiovascular disease, myocardial infarction, stroke, lung cancer, and respiratory diseases such as asthma and all-cause mortality) [6,7,8,9]. Ebenstein et al. reported that air pollution in North China decreased life expectancy by 3.1 years [10].

Sleep is crucial to achieving optimal health and well-being [11]. A lack of sleep has been linked to rises in mortality [12], obesity [13], heart disease [14], diabetes [15], cancer [16], and depression [17]. Sleep deprivation among young adults is rapidly increasing, and it has become a significant health concern in recent years [11]. The Behavioral Risk Factor Surveillance System (BRFSS) reported that more than 34% of adults received on average less than 7 h of sleep per night in 2014 [18]. Environmental factors play a crucial role in sleep quality. Previous studies have reported that specific environmental factors such as traffic noise [19,20,21,22], temperature [23], housing, and living in social neighborhoods [24] are all associated with sleep duration [25].

To our knowledge, only a few studies have examined the relationship between air pollution and sleep duration. Previous studies have focused primarily on the effect of air pollution on sleep-related respiratory symptoms such as sleep-disordered breathing [23,26,27,28], obstructive sleep apnea (OSA), and sleep apnea [29,30]. However, to our knowledge, no study has examined the relationship between ambient air pollution and sleep duration. Studies have been performed to assess the relationship between traffic-related air pollution, i.e., black carbon [BC] and sleep duration [31]. However, levels of air quality index (AQI), PM_2.5_, PM_10_, and NO_2_ were excluded.

This study investigates the associations between air pollution and sleep duration among freshmen in Beijing, China over a five-year study period (2013–2018). The purpose of this study is to estimate the associations between air pollution and sleep duration among freshman students in Beijing, China.

## 2. Methods

### 2.1. Participants

Data from this study came from a paper–pencil based health survey that all freshmen at Tsinghua University are required to take. Participant consent to record survey responses was obtained. Designed to evaluate the participants’ physical and mental health conditions, the detailed descriptions of the questionnaire content, implementation procedures, and member cohorts have been published [32,33].

Data on five cohort groups were collected at Tsinghua University from 2013–2014, 2014–2015, 2015–2016, 2016–2017, and 2017–2018 (*n* = 31,806). The size of each cohort varied slightly: 2013–2014 (*n* = 5979; 66.3% male), 2014–2015 (*n* = 8155; 65.2% male), 2015–2016 (*n* = 5929; 67.9% male), 2016–2017 (*n* = 5305; 71.1% male), and 2017–2018 (*n* = 6438; 68.9% male). A total of 16,889 Tsinghua University freshmen students enrolled and responded to the survey. Among them, 3752 completed the survey only once, 13,137 freshmen participated in more than one survey. The final sample (*n* = 31,582) had non-missing values for the specific outcome and all covariates (see Figure 1).

The study was approved by the Tsinghua University Institutional Review Board (IRB #2012534001).

### 2.2. Sleep Measurement

The present study used the Chinese version of the Pittsburgh Sleep Quality Index (CPSQI) to measure sleep quality. The CPSQI has a test–retest reliability of 0.85 and a validity of 0.64 (by a 7-day daily sleep log) [34]. This study measured sleep duration based on the following question adopted from the CPSQI: “On average, how many hours of sleep did you get per night in the last week?” [34,35,36].

### 2.3. Environmental Measures

Environmental measures included average daytime temperature (°C), average wind speed (m/s), and percentage of rainy days in Beijing, China. These measures were taken seven days before the survey was given. Daytime temperature, wind speed, and percentage of rainy days were made available by the China Meteorological Administration. The following air pollution data and environmental factors were measured—air quality index (AQI), PM_2.5_, PM_10_, and NO_2_. The Beijing Municipal Ecological Environment Bureau provided hourly AQI, PM_2.5_, PM_10_, and NO_2_ data. Data were collected at Wan Liu in the district of Hai Dian, which is approximately 5 km from Tsinghua University. Daily weather data were made available by the China Meteorological Administration and included daytime temperature, wind speed, and percentage of rainy days.

The data on average air pollution concentration were standardized through demeaning (i.e., subtracting the mean from each value) and then dividing by its own standard deviation (i.e., PM_2.5_ z-scores). The estimated coefficient of air pollution concentration can be interpreted as the change in an outcome variable (e.g., total daily hours of sleep in the last week) with respect to a change in air pollution concentration by one standard deviation. Over the five-year study period, air pollution concentrations in Beijing, China fluctuated. The mean and standard deviation (SD) concentrations of AQI, PM_2.5_, PM_10_, and NO_2_ were 104.11 (SD = 77.50), 69.15 µg/m^3^ (SD = 70.99), 106.04 µg/m^3^ (SD = 73.45), and 56.06 µg/m^3^ (SD = 27.39), respectively.

### 2.4. Statistical Analyses

The present study used means, SD, and percentages to summarize and compared for characteristics of the overall sample. Continuous variables were examined using the *t*-test and the Mann–Whitney U test. Categorical variables were collated using chi-square tests. Multilevel mixed-effects linear regression models were performed based on the repeated-measure survey data from the five freshman cohorts (2013–2014, 2014–2015, 2015–2016, 2016–2017, and 2017–2018). The daily average hours of sleep in the past week was used as the study’s continuous outcome variable. The key independent variables of the study were AQI, PM_2.5_, PM_10_, and NO_2_. Environmental measures (i.e., average daytime temperature, average wind speed, and percentage of rainy days) were included in the study’s independent variables. An entire sample with both genders, male only and female only, used separate regression models that included each outcome variable.

We examined the effects of air pollution concentration on individual-level sleep outcomes based on the survey data from five freshman cohorts (2013–2018) at Tsinghua University. The present study used a mixed-effect model containing both fixed effects and random effects. There were also two levels of non-independence (person level and cohort week level) in the model. We chose the random effects estimation method using maximum likelihood (ML). Result are presented as regression coefficients and their 95% confidence intervals (CI), the level of statistical significance used at 0.05. Multilevel mixed-effects linear regression model measures were assessed using the Stata command ‘xtmixed’. Likelihood ratio tests (LR) were used to assess the utility of multilevel modeling against the null model that assumed no clustering within the data structure.

In all regressions, the key independent variable was the standardized average air pollution concentration over the seven days before the survey. All models were included in the individual-level time-variant covariates as well as environmental variables, which included average daytime temperature, average wind speed, and percentage of rainy days over the last seven days.

Individual-level covariates were controlled for in the regression analyses. Continuous variables for age in years, body mass index (BMI; kg/m^2^), self-rated physical health (1–10, poor–excellent), and self-rated mental health (1–10, poor–excellent) were included. Dichotomous variables for current smoking status (non-smokers as the reference group) and drinking status (non-drinkers as the reference group) were additionally measured.

All statistical procedures were performed in Stata 14.2 SE version (StataCorp, College Station, TX, USA).

## 3. Results

### 3.1. The Characteristics of Participants

Table 1 presents baseline characteristics of the survey participants. Males accounted for almost two-thirds (*n* = 21,512; 67.64%) of the total sample (*n* = 31,806). The mean age in years was 18.4 (SD = 0.9). The percentage of the total sample size by cohort is as follows: 2013–2014 (*n* = 5979; 18.80%), 2014–2015 (*n* = 8155; 25.64%), 2015–2016 (*n* = 5929; 18.64%), 2016–2017 (*n* = 5305; 16.88%), and 2017–2018 (*n* = 6438; 20.24%). The mean BMI was 21.4 kg/m^2^ (SD = 3.3). Only 0.5% of participants were current smokers and 2.7% were current drinkers. Self-rated physiological health and self-rated psychological health scores had mean 5.34 (SD = 2.19) and 6.23 (SD = 2.45), respectively.

### 3.2. The Sleep Variations

Table 2 shows the mean daily hours of sleep in the last week in the survey. The mean daily average hours of sleep by cohort were as follows: 7.09 h (SD = 0.90) in 2013–14 cohort wave 1; 7.27 h (SD = 0.97 in 2013–2014 cohort wave 2; 7.03 h (SD = 0.78) in 2014–2015 cohort wave 1; 7.97 h (SD = 1.21) in 2014–2015 cohort wave 2; 7.43 h (SD = 1.15) in 2014–2015 cohort wave 3; 7.03 h (SD = 0.73) in 2015–2016 cohort wave 1; 7.35 h (SD = 1.39) in 2015–16 cohort wave 2; 7.13 h (SD = 0.93) in 2016–2017 cohort wave 1; 7.19 h (SD = 0.98) in 2015–2016 cohort wave 2; 6.99 h (SD = 1.01) in 2017–2018 cohort wave 1; 7.15 h (SD = 1.19) in 2017–2018 cohort wave 2.

### 3.3. The Air Pollution Variations

Table 3 shows the mean variations of air pollution in the surveys. Large variations in air pollution occurred. For example, AQI ranged from 67.71 to 234.00 during the 2014–2015 cohort. Figure 2 shows the variations of AQI and sleep in the follow–up surveys. It demonstrates there was a declining outcomes trend in sleep in last week with AQI air pollution, increasing among the freshmen over the survey period.

### 3.4. Association of Air Pollution with Sleep

Table 4 shows the estimated effects of air pollution concentration on individual-level outcomes on sleep using multilevel mixed-effects linear regressions. Air pollution concentration was found to be negatively associated with sleep duration among survey participants. Specifically, a one standard deviation (SD) increase in AQI (77.50) was associated with a reduction in total daily hours of sleep by 0.68 (95% confidence interval (CI) = 0.63, 0.73).

An increase in particulate matter concentration in PM_2.5_ and PM_10_ by one SD (i.e., PM_2.5_ by 70.99 µg/m^3^ and PM_10_ by 73.45 µg/m^3^) was associated with a reduction in daily hours of sleep by 0.55 (95% CI = 0.51, 0.59) and 0.70 (95% CI = 0.64, 0.76), respectively.

An increase in gaseous pollutant matter concentration in NO_2_ by one SD (NO_2_ by 28.68 µg/m^3^) was associated with a reduction in daily hours of sleep by 0.51 (95% CI = 0.47, 0.54).

## 4. Discussion

The purpose of our study was to estimate the relationship between air pollution and sleep duration among freshman students. This study investigated the associations between air pollution concentration and daily sleep duration among university freshmen in Beijing, China over five years (2013–2018). The air pollution level was found to be negatively linked to sleep duration among our survey participants. To our best knowledge, this is the first study to reveal that an increase in air pollution is associated with a reduction in sleep duration in youth. Our finding builds on previous research done on the associations between air pollution and health behaviors and highlights the associations of air pollution on sleep duration.

In our study, air pollution was negatively associated with sleep duration among our participants. Our results suggested that a one SD increase in AQI (77.50) was associated with a reduction in sleep duration by 0.68 h per day. Our findings are consistent with previous research [31,37] that reported air pollution to have a negative impact on sleep among adults and children. Fang et al. found that an increase in annual black carbon (BC) (0.21 µg/m^3^) was associated with 0.23–0.25 fewer hours of sleep among American adults aged 53.8 years old on average [31].

Our findings are also consistent with previous research [23,26,28,38], which found that increasing levels of ozone and PM_10_ were associated with a reduction in sleep efficiency [23], an increase in sleep–disordered breathing (SDB) [23,26,28], and an increase in the severity of sleep apnea [38]. In a recent study across seven northeastern Chinese cities, higher long-term exposure to PM_2.5_, PM_10_, SO_2_, NO_2,_ and O_3_ was associated with a 1.02–1.53 greater odds ratio (OR) of sleep disorders in Chinese children [39]. Moreover, a separate study reported that prenatal PM_2.5_ exposure was associated with altered sleep in preschool-aged children in Mexico City [40]. These prior studies, however, studied the relationship between air pollution and sleep disorders in other populations. This present research explicitly focused on sleep duration among Chinese youth.

There are several potential pathways in which air pollution may decrease sleep duration. Air pollution may alter sleep through effects on the central nervous system [41] by translocating particles from the air to the nose up through the olfactory nerve into the brain, striatum frontal cortex, and cerebellum [42,43]. This, in turn, is linked with increased brain inflammatory responses [44] and changes in neuro-transmitter levels [23,45]. Air pollution may also reduce sleep duration by influencing the central ventilatory control centers [23]. Particulate matter particles may increase the risk for nasal or pharyngeal inflammatory responses, which increase upper airway resistance and reduce airway patency. Partial obstruction to the airway may cause hypoxia in the tissues and decrease sleep duration [23,38,46]. Air pollution may also reduce sleep duration by mental function. Increases in air pollution may cause increases in depression and anxiety [47,48,49]. However, research is needed to further understand the role of these potential pathways on sleep duration.

Overall, the results of this study build on our previous research done on the association of air pollution with health behaviors—physical activity [32], sedentary behavior [33]—among Chinese freshman students. Two of our past studies demonstrate that increased levels of air pollution are linked to reduced physical activity [32] and increased sedentary behavior [33]. These findings are consistent with our earlier study among older Chinese adults, specifically university retirees, that reported increased levels of air pollution reduced physical activity [50]. Additionally, our previous findings among Chinese freshmen are consistent with those of other Chinese studies that reported increased levels of AQI are associated with a reduction in physical activity and an increase in sedentary behavior among Chinese adults [51,52]. Findings of other studies found an association between air pollution exposure and sedentary behavior [50], while this present study analyzed the association on sleep duration [50].

This is the first study to use a large cohort sample of freshmen at Tsinghua University from 2013 to 2018 to investigate the association between air pollution and sleep duration. In addition, the study used a follow-up study design and reliable and time-sensitive environmental measurements. However, there are a few major limitations to this study that should be noted. First, the generalizability of these results is subject to certain limitations. All freshman cohorts were students at Tsinghua University; therefore, the results are unlikely to represent the entire undergraduate population in Beijing or nationwide. Second, although there is a large sample size, the study was limited to young adults aged 18 to 19 years. It is not possible to apply the results of this study to middle-aged and older adults. In addition, we did not collect data on time spent each day studying. In the future, considerable work will need to be done to determine whether this specific modeling can be employed in other regionally or nationally representative samples.

Further research using representative samples would help us to establish a greater degree of generalizability on air pollution exposure. We used ambient (outdoor) air pollution instead of indoor air pollution data to examine the relationship between air pollution and sleep duration. Sleep disturbances may be more sensitive to indoor air pollution. A future study with indoor air pollution assessment is needed. An additional significant confounder and limitation to the present study are any seasonal variations in air pollution.

We did not use an objective approach to measure sleep duration in this study but rather a subjective self-report to represent the total daily hours of sleep. While we adopted a standardized well-validated and reliable questionnaire, the Chinese version of the Pittsburgh Sleep Quality Index (CPSQI) [53,54], self-reported sleep duration is subject to recall error and social desirability bias [55]. A future study with objectively measured sleep assessment (e.g., wrist actigraphy [56], sleep-electroencephalography [57], or polysomnography [58]) is therefore suggested. Influences on individual sleep duration are complex and may include more than just air pollution levels. The Chinese version of the the Pittsburgh Sleep Quality Index (CPSQI) question on sleep duration asks only about sleep over the past week; therefore, more specific air quality data could not be collected. Non-independence would also need to be modeled. The use of air quality over the seven days prior to the questionnaire being completed means that the association being investigated does not have any lags and cannot be specific to single nights. Other demographics such as environmental and socioeconomic factors may play a key role in determining sleep duration.

## 5. Conclusions

This study investigated the associations between air pollution and sleep duration among university freshmen in Beijing, China. The results of this investigation show that, a one standard deviation (SD) increase in environmental variables, i.e., AQI, PM_2.5_, PM_10_, and NO_2_, may be associated with a reduction in sleep duration by 30 min or more. A key policy priority should, therefore, be to plan for the long-term care of air pollution in China. Future research in other cities and universities is needed to fully understand the association between air pollution and sleep.

## Figures and Tables

**Figure 1 ijerph-16-03362-f001:**
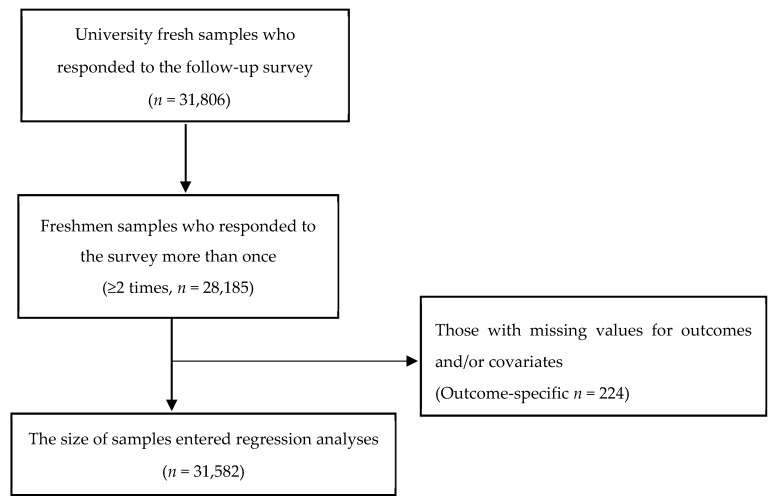
Study sample flowchart.

**Figure 2 ijerph-16-03362-f002:**
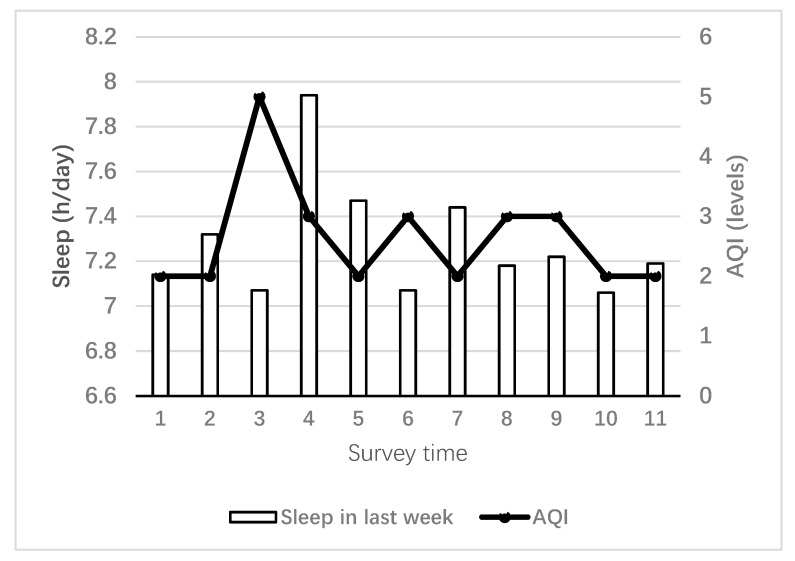
Trend for sleep in last week and air quality index (AQI) concentration among the freshman cohort. Survey time 1: 2013–2014 cohort (9–15 Dec); 2: 2013–2014 cohort (5–11 May); 3: 2014–2015 cohort (6–12 Oct); 4: 2014–2015 cohort (24 Feb–2 Mar); 5: 2014–2015 cohort (4–10 May); 6: 2015–2016 cohort (14–20 Sep); 7: 2015–2016 cohort (2–8 May); 8: 2016–2017 cohort (21–27 Nov); 9: 2016–2017 cohort (15–21 May); 10: 2017–2018 cohort (13–19 Nov); 11: 2017–2018 cohort (30 Apr–6 May).

**Table 1 ijerph-16-03362-t001:** The characteristics of survey participants.

Characteristics	Male	Female	Total	*p*
Sex, *n* (%)	21,512 (67.64)	10,294 (32.36)	31,806	
^a^ Age (year), mean (SD)	18.46 (0.90)	18.40 (0.81)	18.44 (0.87)	0.001
^c^ Freshman cohort, *n* (%)				
2013–2014	3966 (66.33)	2013 (33.67)	5979 (18.80)	0.001
2014–2015	5322 (65.26)	2833 (34.74)	8155 (25.64)	
2015–2016	4026 (67.90)	1903 (32.10)	5929 (18.64)	
2016–2017	3770 (71.07)	1535 (28.93)	5305 (16.68)	
2017–2018	4428 (68.78)	2010 (31.22)	6438 (20.24)	
^b^ Body mass index, mean (SD)				
BMI (kg/m^2^)	21.89 (3.18)	20.35 (2.77)	21.40 (3.14)	0.001
^a^ Smoking, *n* (%)	127 (84.67)	23 (15.33)	159 (0.47)	0.001
^a^ Drinking, *n* (%)	714 (82.83)	148 (117.17)	862 (2.69)	0.001
^b^ Self-rated physical health, mean (SD)				
Physical health score (1–10)	5.19 (2.35)	5.08 (2.29)	5.16 (2.33)	0.001
^b^ Self-rated mental health, mean (SD)				
Mental health score (1–10)	6.07 (2.63)	5.97 (2.63)	6.04 (2.63)	0.001
^a^ Disease number, mean (SD)	0.57 (0.49)	0.62 (0.49)	0.59 (0.49)	0.001

Note: ^a^
*p*-values come from the Mann–Whitney U test; ^b^
*p*-values come from *t*-tests; ^c^
*p*-values come from Chi-squared tests.

**Table 2 ijerph-16-03362-t002:** Mean hours of sleep in the last week (h/week) (*n* = 31,806).

Characteristics	*n*	Male	Female	Total	*p*
2013–2014 cohort (9–15 Dec)	3009	7.14 (0.88)	6.99 (0.93)	7.09 (0.90)	0.001
2013–2014 cohort (5–11 May)	2970	7.32 (0.95)	7.17 (0.99)	7.27 (0.97)	0.001
2014–2015 cohort (6–12 Oct)	2996	7.07 (0.77)	6.94 (0.78)	7.03 (0.78)	0.001
2014–2015 cohort (24 Feb–2 Mar)	3002	7.94 (1.12)	8.01 (1.35)	7.97 (1.21)	0.605
2014–2015 cohort (4–10 May)	2157	7.47 (1.14)	7.34 (1.15)	7.43 (1.15)	0.001
2015–2016 cohort (14–20 Sep)	2922	7.07 (0.71)	6.94 (0.76)	7.03 (0.73)	0.001
2015–2016 cohort (2–8 May)	3007	7.44 (1.50)	7.18 (1.09)	7.35 (1.39)	0.001
2016–2017 cohort (21–27 Nov)	3192	7.18 (0.93)	7.02 (0.94)	7.13 (0.93)	0.001
2016–2017 cohort (15–21 May)	2113	7.22 (0.97)	7.12 (1.03)	7.19 (0.98)	0.038
2017–2018 cohort (13–19 Nov)	3056	7.06 (1.01)	6.84 (1.00)	6.99 (1.01)	0.001
2017–2018 cohort (30 Apr–6 May)	3382	7.19 (1.09)	7.05 (1.37)	7.15 (1.19)	0.001

Note: *p*-values come from *t*-tests.

**Table 3 ijerph-16-03362-t003:** Average air pollution concentrations and other environmental variables in the last seven days before survey (*n* = 31,806).

Freshman Cohort	2013–2014 Cohort (*n* = 5979) ^a^		2014–2015 Cohort (*n* = 8155) ^b^			2015–2016 Cohort (*n* = 5929) ^c^		2016–2017 Cohort (*n* = 5305) ^d^		2017–2018 Cohort (*n* = 6438) ^e^	
Survey Order	9–15 Dec	5–11 May	6–12 Oct	24 Feb–2 Mar	4–10 May	14–20 Sep	2–8 May	21–27 Nov	15–21 May	13–19 Nov	30 Apr–6 May
AQI	66.43 (30.59)	85.85 (26.31)	234.00 (145.86)	104.71 (67.98)	67.71 (11.63)	116.00 (51.36)	76.57 (40.97)	115.13 (108.73)	120.51 (46.24)	61.96 (28.25)	96.35 (48.43)
PM_2.5_ (µg/m³)	45.00 (26.28)	59.14 (27.52)	194.29 (143.70)	72.29 (57.49)	41.47 (13.16)	87.14 (41.54)	39.57 (17.42)	85.85 (101.17)	61.86 (22.72)	36.11 (27.02)	38.23 (23.61)
PM_10_ (µg/m³)	65.71 (34.52)	95.29 (30.98)	194.43 (128.11)	103.29 (76.98)	79.57 (26.37)	109.28 (39.31)	80.71 (52.10)	115.06 (123.85)	120.49 (32.38)	70.38(32.72)	132.13 (77.78)
NO_2_ (µg/m³)	34.43 (10.49)	61.14 (13.31)	103.00 (39.24)	61.28 (24.42)	40.00 (9.92)	63.29 (13.78)	44.43 (17.82)	64.10 (45.05)	49.28 (22.91)	49.07 (23.43)	46.61 (17.86)
*Covariate variables*											
Temperature (°C)	5.43 (1.40)	20.71 (3.09)	20.86 (1.86)	6.29 (3.45)	21.43 (5.16)	27.00 (0.82)	25.57 (2.44)	(5.57) (1.72)	20.86 (5.18)	2.29 (2.21)	19.43 (1.99)
Wind (m/s)	3.57 (0.67)	3.14 (0.24)	3.43 (0.73)	3.79 (0.91)	3.01 (0.19)	3.00 (0.00)	3.14 (0.24)	3.29 (0.06)	3.15 (0.24)	3.71 (0.95)	3.35 (0.56)
Rain (%)	0.00 (0.00)	0.43 (0.54)	0.14 (0.38)	0.14 (0.38)	0.57 (0.54)	0.14 (0.38)	0.29 (0.49)	0.00 (1.00)	0.14 (0.38)	0.00 (0.00)	0.00 (0.00)

Air pollution data collection periods ^a^ 2013–2014 cohort: 9–15 December 2013, 5–11 May 2014; ^b^ 2014–2015 cohort: 6–12 October 2014, 24 February–2 March 2015, 4–10 May 2015; ^c^ 2015–2016 cohort: 14–20 September 2015, 2–8 May 2016. ^d^ 2016–2017 cohort: 21–27 November 2016, 15–21 May 2017. ^e^ 2017–2018 cohort: 13–19 November 2017, 30 April–6 May 2018.

**Table 4 ijerph-16-03362-t004:** Estimated effects of air pollution on individual-level sleeping outcomes in the past week per day by gender.

Dependent Variable	Male Only		Female Only		Total	
	Coefficient (95% CI)	# Observations	Coefficient (95% CI)	# Observations	Coefficient (95% CI)	# Observations
AQI						
Sleep in last week (h/day)	−0.59 *** (–0.65, −0.53)	21,375	−0.86 *** (–0.94, −0.77)	10,207	−0.68 *** (–0.73, −0.63)	31,582
PM_2.5_						
Sleep in last week (h/day)	−0.49 *** (–0.53, −0.45)	21,375	−0.67 *** (–0.73, −0.61)	10,207	−0.55 ** (–0.59, −0.51)	31,582
PM_10_						
Sleep in last week (h/day)	−0.59 ** (–0.66 −0.52)	21,375	−0.96 *** (–1.07, −0.84)	10,207	−0.70 *** (–0.76, −0.64)	31,582
NO_2_						
Sleep in last week (h/day)	−0.44 *** (–0.48, −0.39)	21,375	−0.68 *** (–0.75, −0.61)	10,207	−0.51 *** (–0.54, −0.47)	31,582

Note: Multilevel mixed-effects linear regression models were performed to analyze the effects of air pollution concentrations on participants stratified by gender. Models adjust for covariates listed in Table 1 (i.e., smoking status, age, BMI, drinking, self-rated physical health and mental health) and environmental variables listed in Table 2 (i.e., temperature and rainy days in the last week). There were 16,870 participants with 11,452 male participants and 5538 female participants in the regression. * *p* < 0.05; ** *p* < 0.01; *** *p* < 0.001.

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
