# Peer review of "The Association between Air Pollution and Sleep Duration: A Cohort Study of Freshmen at a University in Beijing, China"

_ijerph, 2019, doi:10.3390/ijerph16183362_

Round 1
Reviewer 1 Report
I still think the large variability of air pollution values need to be discussed more as exposure assessment is a key uncertainty in this study.
Author Response
Responses to Reviewer 1:
Point 1. I still think the large variability of air pollution values need to be discussed more as exposure assessment is a key uncertainty in this study.
Response: We thank the reviewer for the comments. We have revised the discussion as suggested. See Line 243-248. Again, thank you very much!
Reviewer 2 Report
I think much more care is needed around causal language. The use of “impact” (title; Lines 17, 66, etc.), “decreases” (Line 29), and “leads to” (Line 200), as examples, suggest causal links rather than observed associations and I strong recommend the authors work through their manuscript and reword any causal language into appropriate association language. As part of this, I’d also suggest replacing references to increases/decreases with higher/lower as you are not looking at changes but rather cross-sectional associations. The standardisation process on Lines 102–106, unless I am misunderstanding the text here, simply makes the variables relative to their overall mean and does not make it about “changes”. I do not see this as a longitudinal study (Lines 64, 131, 196, 248, and 273) in terms of the statistical analyses as there is no temporal component to the models despite the longitudinal data. The analyses consist of looking at cross-sectional associations with standard errors adjusted for repeated measures on the same individuals.
A substantial challenge for a study of this type is the matching of the air quality data to the sleep quality data in an appropriate manner. If sleep diaries or accelerometry, for example, had been used, it would be possible to match air quality, over some appropriate period, with each night’s sleep. The standard PSQI asks about sleep over the past month, so cannot be used with more finely grained air quality data, although based on Line 90, it appears that the question adapted from the Chinese version is based on the previous week only and so air quality data could be over the same one week period (as seems to be the case). I’m not aware of the PSQI being used for a one week period, which raises questions about validity and reliability which should be addressed in the manuscript. In any case, this would create one-week bins within which the error terms would be potentially correlated (due to unmodelled shared exposures) and this non-independence would also need to be modelled. The use of air quality over the 7 days prior to the questionnaire being completed (Lines 127—and for other independent variables Line 129) means that the association being investigated does not have any lags and cannot be specific to single nights (the effect in this case would be diluted over the week’s data). This is entirely reasonable if the mechanism is assumed to be immediate, not cumulative or delayed, and not specific to individual nights, but I think this point needs justification. I am assuming here that each cohort consists of students who completed their questionnaire in different weeks to other students. Based on Table 2, it appears that there are 11 such cohort-weeks in the study. This information in Table 2 would be have been very useful to have in the methods section.
It’s not clear to me why only students with multiple cross-sectional data were included in analyses (students with only one set of data would still inform the association) but the clustering within individuals (along with cohort weeks as mentioned above) needs to be accommodated through random effects or in some other manner for all analyses that use more than one cohort. The cohort-week bins would still potentially create non-independence of the error terms within cohorts (with 11 levels for this) and so two random effects, or equivalent, would be needed in these cases.
The approach used here is not what I would call a “fixed effects” approach to longitudinal data. In this context, with repeated measures, this suggests to me that student ID was included in the model (this would be the fixed effect which warrants mention), which would make the results non-generalisable outside of these students (a reason for using random effects) and would not adjust the denominator degrees of freedom appropriately. Based on Line 154, it appears that standard errors were clustered within individuals, but this approach does not, and cannot be extended to, address the cohort-week clustering. This might simply be a terminology issue, but I think the approach to handling the clustering within individuals needs to be made clear earlier in the statistical methods. Lines 268–269 seem to be specifically referring to the inclusion of participant IDs as a variable in the statistical models.
I’m afraid that, some of which I assume to have been added to justify the modelling approach taken, Lines 131–146 do not make much sense to me as a biostatistician. Fixed effects models are not only not widely accepted for analysis of cohorts or longitudinal data (Line 131), the opposite is the case. Note that a fixed effects model approach here would, as noted above for standard use of the term in this context, include participant ID as an independent variable in the model, which does not appear to be the case here—if this was done alongside robust clustered standard errors, the estimated standard errors would be nonsensical. This, however, does not correctly adjust the denominator degrees of freedom. If participant IDs are not included, then the arguments against adjusting for gender and ethnicity (Lines 141–142 and 145) do not make sense as including these could still reduce the error variance and Lines 268–269 are not correct.
Correct analyses of this data could include linear mixed models, there are two levels of non-independence (person-level and cohort week-level) precluding some other approaches, or a summary measures approach collapsing data down to independent statistics (e.g. mean sleep and air quality for each cohort week and using only the first questionnaire for each student in calculating these means) which I think would give only 11 rows of data here. At best, the current approach addresses within person-correlations through adjusting the standard errors (which does not match Lines 133–146 and Lines 268–269 which talks as if a participant ID is included as a fixed effect) but it does not address the temporal repeated measures in the 11 cohort weeks. Two participants completing their questionnaire at the same time might have their past week’s sleep disturbed by some shared factor (e.g. exams or night-time noise) and so are not independent of one another in the same way that two participants from different cohort weeks would be. Note also that the argument on Lines 133–146 would not support excluding participants due to missing non-varying characteristics, including “sex, race, body height” (Line 80).
If claims about female students having a different association than male students (Lines 27–28 and Lines 200–201) are to be made, this needs to be tested formally through using an interaction to assess evidence for effect modification. Stratified analyses does not permit claiming differences in associations between the strata.
I would have presented the participant characteristics (Table 1) for each cohort in case there were changes over time. Even better would be broken down by cohort-weeks as you do for air quality in Table 2.
There are various issues with the writing and careful proofreading is needed throughout the manuscript to ensure that expression is clear and uses standard English.
Author Response
Responses to Reviewer 2:
Point 1. I think much more care is needed around causal language. The use of “impact” (title; Lines 17, 66, etc.), “decreases” (Line 29), and “leads to” (Line 200), as examples, suggest causal links rather than observed associations and I strong recommend the authors work through their manuscript and reword any causal language into appropriate association language. As part of this, I’d also suggest replacing references to increases/decreases with higher/lower as you are not looking at changes but rather cross-sectional associations. The standardisation process on Lines 102–106, unless I am misunderstanding the text here, simply makes the variables relative to their overall mean and does not make it about “changes”. I do not see this as a longitudinal study (Lines 64, 131, 196, 248, and 273) in terms of the statistical analyses as there is no temporal component to the models despite the longitudinal data. The analyses consist of looking at cross-sectional associations with standard errors adjusted for repeated measures on the same individuals.
Response: We want to express our gratitude to the reviewer 2’s meaningful comments. We have taken the comments/suggestions very seriously and have made all corresponding modifications as suggested. We have revised based on the comments. We used “associations” instead of “impact”, “reduction” instead of “decrease”, and “was associate with” instead of “lead to”. We agree with her/his comments and modified our study to use appropriate associative language. Thank you for the note.
Point 2. A substantial challenge for a study of this type is the matching of the air quality data to the sleep quality data in an appropriate manner. If sleep diaries or accelerometry, for example, had been used, it would be possible to match air quality, over some appropriate period, with each night’s sleep. The standard PSQI asks about sleep over the past month, so cannot be used with more finely grained air quality data, although based on Line 90, it appears that the question adapted from the Chinese version is based on the previous week only and so air quality data could be over the same one week period (as seems to be the case). I’m not aware of the PSQI being used for a one week period, which raises questions about validity and reliability which should be addressed in the manuscript. In any case, this would create one-week bins within which the error terms would be potentially correlated (due to unmodelled shared exposures) and this non-independence would also need to be modelled. The use of air quality over the 7 days prior to the questionnaire being completed (Lines 127—and for other independent variables Line 129) means that the association being investigated does not have any lags and cannot be specific to single nights (the effect in this case would be diluted over the week’s data). This is entirely reasonable if the mechanism is assumed to be immediate, not cumulative or delayed, and not specific to individual nights, but I think this point needs justification. I am assuming here that each cohort consists of students who completed their questionnaire in different weeks to other students. Based on Table 2, it appears that there are 11 such cohort-weeks in the study. This information in Table 2 would be have been very useful to have in the methods section.
Response: Thank you for the suggestions. We addressed the limitation of self-reporting the seven day measurement of sleep quality in PSQI in the limitation part. See Line 250-261. We added the test-retest reliability of 0.85 and validity of 0.64 of the Chinese version of the Pittsburgh Sleep Quality Index (CPSQI). See Line 87-88.
Point 3. It’s not clear to me why only students with multiple cross-sectional data were included in analyses (students with only one set of data would still inform the association) but the clustering within individuals (along with cohort weeks as mentioned above) needs to be accommodated through random effects or in some other manner for all analyses that use more than one cohort. The cohort-week bins would still potentially create non-independence of the error terms within cohorts (with 11 levels for this) and so two random effects, or equivalent, would be needed in these cases.
Response: Thank you for the suggestions. We have added data on freshmen who completed survey only once data into the mixed effect model. The final sample (n = 31,582) had non-missing values for the specific outcome and all covariates. (see Figure 1).
Point 4. The approach used here is not what I would call a “fixed effects” approach to longitudinal data. In this context, with repeated measures, this suggests to me that student ID was included in the model (this would be the fixed effect which warrants mention), which would make the results non-generalisable outside of these students (a reason for using random effects) and would not adjust the denominator degrees of freedom appropriately. Based on Line 154, it appears that standard errors were clustered within individuals, but this approach does not, and cannot be extended to, address the cohort-week clustering. This might simply be a terminology issue, but I think the approach to handling the clustering within individuals needs to be made clear earlier in the statistical methods. Lines 268–269 seem to be specifically referring to the inclusion of participant IDs as a variable in the statistical models.
Response: We have revised our manuscript as suggested. We have used a mixed-effects model (random+fixed effects) in the statistical methods. See Line 114-116, Line 122-127.
Point 5. I’m afraid that, some of which I assume to have been added to justify the modelling approach taken, Lines 131–146 do not make much sense to me as a biostatistician. Fixed effects models are not only not widely accepted for analysis of cohorts or longitudinal data (Line 131), the opposite is the case. Note that a fixed effects model approach here would, as noted above for standard use of the term in this context, include participant ID as an independent variable in the model, which does not appear to be the case here—if this was done alongside robust clustered standard errors, the estimated standard errors would be nonsensical. This, however, does not correctly adjust the denominator degrees of freedom. If participant IDs are not included, then the arguments against adjusting for gender and ethnicity (Lines 141–142 and 145) do not make sense as including these could still reduce the error variance and Lines 268–269 are not correct.
Response: We thank Reviewer 2 for the comment. We agree with the comment and the change has been made as suggested in the statistical analyses section and the discussion section. See Line 122-127.
Point 6. Correct analyses of this data could include linear mixed models, there are two levels of non-independence (person-level and cohort week-level) precluding some other approaches, or a summary measures approach collapsing data down to independent statistics (e.g. mean sleep and air quality for each cohort week and using only the first questionnaire for each student in calculating these means) which I think would give only 11 rows of data here. At best, the current approach addresses within person-correlations through adjusting the standard errors (which does not match Lines 133–146 and Lines 268–269 which talks as if a participant ID is included as a fixed effect) but it does not address the temporal repeated measures in the 11 cohort weeks. Two participants completing their questionnaire at the same time might have their past week’s sleep disturbed by some shared factor (e.g. exams or night-time noise) and so are not independent of one another in the same way that two participants from different cohort weeks would be. Note also that the argument on Lines 133–146 would not support excluding participants due to missing non-varying characteristics, including “sex, race, body height” (Line 80).
Response:
Thank you for the suggestions. We have revised as suggested. We used multilevel mixed-effects linear regression models instead of fixed-effect liner regression model to analyze this data. See Line 122-127.
Point 7. If claims about female students having a different association than male students (Lines 27–28 and Lines 200–201) are to be made, this needs to be tested formally through using an interaction to assess evidence for effect modification. Stratified analyses does not permit claiming differences in associations between the strata.
Response: Thank you for the suggestion. The changes have been made as suggested. We have deleted claims about female students having a different association than male students in the manuscript. See Lines 27-28. Line 189-190. Line 220.
Point 8. I would have presented the participant characteristics (Table 1) for each cohort in case there were changes over time. Even better would be broken down by cohort-weeks as you do for air quality in Table 2.
Response: Thank you for the suggestion. The changes have been made as suggested. See table 1. Line 149-155.
Point 9.There are various issues with the writing and careful proofreading is needed throughout the manuscript to ensure that expression is clear and uses standard English.
Response: We have invited our coauthor, Shelby Paige Gordon, a native English speaker to proofread the manuscript. We have significantly re-written the manuscript as suggested. We believe the readability of this manuscript is better.
We want to take this opportunity to thank Reviewer #2 for providing the detailed and helpful comments/suggestions for the manuscript.
Reviewer 3 Report
The authors examined the association between air polution and sleep duration.
major comments:
- There are a lot of English grammar and spelling errors. Please rephrase. I note here a few, but there a many more in the text. To list a few:
line 36-37: 2 times "in 2018" in the sentence.
line 53: "environment" should be "environmental"
line 60: "none" should be "no"
line 61: "assessed" should be "assess"
my main methodological comment is that the authors present this work as a longitudinal study, but in fact the data that is presented in this study is based on a cross-sectionial analysis based on repeated measures (which takes into account the correlation between the cross-sectional assessments). It the authors really want to study the longitudinal effect, which is possible with the data they have, they should specifically look at whether a change in air polution between 2 survey moments is associated with a change in sleep duration. For example, is there an association between an increase in air polution and a shortening in the sleep duration. The authors nicely present the seasonal variation in air polution. Is there also a seasonal variation in sleep duration? The authors state that women show a significant stronger effect of air polution on the sleep duration than men, but they do not formally test this using a multiplicative interaction term in the statistical analyses.
Author Response
Responses to Reviewer 3:
Point 1. major comments:
- There are a lot of English grammar and spelling errors. Please rephrase. I note here a few, but there a many more in the text. To list a few:
line 36-37: 2 times "in 2018" in the sentence.
line 53: "environment" should be "environmental"
line 60: "none" should be "no"
line 61: "assessed" should be "assess"
Response: Thank you for the suggestion. Thanks for the feedback. We have invited a native English speaker to proofread the manuscript. We believe the readability of this manuscript has improved.
Point 2.
My main methodological comment is that the authors present this work as a longitudinal study, but in fact the data that is presented in this study is based on a cross-sectionial analysis based on repeated measures (which takes into account the correlation between the cross-sectional assessments). It the authors really want to study the longitudinal effect, which is possible with the data they have, they should specifically look at whether a change in air polution between 2 survey moments is associated with a change in sleep duration. For example, is there an association between an increase in air polution and a shortening in the sleep duration. The authors nicely present the seasonal variation in air polution. Is there also a seasonal variation in sleep duration?
Response: Thank you for the meaningful comments. The changes have been made as suggested. We changed causal language into appropriate assocoative language. We changed our title “Impact of Air Pollution on Sleep Duration: A Cohort Study of Freshmen at a University in Beijing, China” to “The Association between Air Pollution and Sleep Duration: A Cohort Study of Freshmen at a University in Beijing, China”. We also used mixed effects models (random+fixed effects) instead of fixed-effect models in the statistical methods. See Line 114-127. We also have added sleep data in table 1. See table 1 and Line 149-155.
Point 3. The authors state that women show a significant stronger effect of air polution on the sleep duration than men, but they do not formally test this using a multiplicative interaction term in the statistical analyses.
Response: Thank you for the comments. We have deleted claims about female students having a different association than male students in the manuscript. See Lines 28, Line 189. See Line 220. Again, thank you very much!
Round 2
Reviewer 2 Report
Thank you for your constructive responses to my comments. The manuscript reads much better now.
There are still quite a few language issues and I’ll note some of these below among my specific comments. I will inevitable have missed some of these issues in those comments though and careful proofreading will still be required to identify them.
First, though, unfortunately, the modelling used here is still not at all clear to me. The only sentence about the statistical methods in the abstract is Line 25: “Linear individual fixed-effect regressions were used to analyze the data.” which would suggest that analyses were conducted at the individual level and were fixed effects only, which wouldn’t make sense. Line 163 repeats this reference to “linear individual fixed-effect regressions”. Back in the statistical methods, Lines 113–114 refers to RM-ANOVA but I cannot see where these were used (or think why they would be needed). (Note also that the t-tests on Line 114 make assumptions about normality and, in its default form, homogeneity of variance for residuals and these diagnostics need to be described here to reassure the reader. Based on data in Table 1, some of the models will not pass diagnostic checks and alternative approaches will be needed such as Mann-Whitney U.) Lines 114–115 and 124–126 do not make it explicit what the random effects are (even if they imply what they might be). Given this description and the data, I don’t think it would be obvious to a (bio)statistician what you have done here and further explanation is definitely needed. My suggestion would be to include sufficient information so that if I, or another (bio)statistician, had your data and the statistical methods text, it would be possible to replicate your results without needing to make guesses or take a trial and error approach. The best way to do this, in my opinion, would be to include Stata code as an online supplement as this is unambiguous and complete. The final sentence here (Lines 126–127) could be deleted or reworded to note that such an approach is required (not simply useful) in the presence of hierarchical or clustered data. In the discussion, on Lines 257–259, you restate some of the points I made last time but this leaves me confused as to what you actually did. I suspect that a very clear statement of the models, including random effects (where these are explicitly described) and/or Stata code will help me and other readers to understand the models used.
Related to this, please also indicate any specific options chosen, including the random effects estimation approach (e.g. REML). As with the t-tests, model diagnostics for the mixed effects models need to be described for the reader’s reassurance, here including normality (for residuals and BLUPs), homoscedasticity, and linearity. The last of these could be particularly important as the association could be rapid growth to an asymptote, sigmoidal, or exponential. Finally, it is traditional to note the level of statistical significance used along with a statement for how and why multiplicity was addressed or not addressed when looking at multiple models.
Just as a suggestion for you to consider, I wonder if you could add a figure showing the means for each exposure and the means for sleep duration for each cohort (basically a pictorial representation of the overall column from the bottom of Table 1 and some rows from Table 2). This might help to give readers some idea whether the findings are likely to be from ecological confounding or reflect an actual exposure-outcome relationship. This could even allow you to replace some of the tables with figures. I’ll leave this as a suggestion for you to decide on, perhaps with input from the editor.
Note that Lines 172–173: “The estimated associations of air pollution on individual-level sleep outcomes proved there to be a more significant reduction in sleep duration among females when compared to males.” would require interaction p-values to support unless it was qualified to the much weaker claim about effect sizes being numerically (and without claiming or suggesting that this is statistically significantly) greater in one compared to the other.
Did you consider looking at all pollution indices in the same model. Lines 194–196 invite the reader to wonder if there is one or several pollution indices that explain the association with correlations between the indices leading to the pattern of statistically significant results.
Line 23: “...was gathered…” should be “…were gathered…” (plural to match “levels” on Line 22).
Lines 27–28: As you are talking about decreases in absolute terms, the CI limits should go from smaller in absolute value to larger in absolute value (i.e. “…by 0.68 (95% confidence interval [CI] = 0.63, 0.73)”. Related to this, on Line 166, the CI limits for the decrease don’t need negative signs as the direction is also made clear by “reduction” on Line 165. See also Lines 169 and 171.
Line 36: “estimates” would be better here than “approximates”.
Line 42: The exponent “3” isn’t superscript (as it is on Line 38).
Lines 50–51: I’m not sure how it can be both “a significant health concern” and “a public health epidemic”. An epidemic in this non-communicable sense is more serious than a significant health concern to me.
Line 88: “test-restes reliability” should be “test-retest reliability”.
Line 88: You need to be specific about what the validity of 0.64 represents (internal consistency through alpha, correlation with a gold-standard measure, etc.).
Lines 107–109: But keep in mind that these are highly positively skewed measurements, as indicated by the SD relative to the mean, and so don’t capture “typical” very well, although they do capture cumulative exposure. You could consider, if you wanted, adding median values to these to indicate “typical” potential exposures. The same issue arises in Table 2.
Line 112: This isn’t a sentence (“Means, SD, and percentages summarized and compared for characteristics of the overall sample.”)
Line 119: Rather than “controlled for”, I think you might mean “were included in”.
Lines 120–121: This sentence needs some attention (including for consistent hyphenation and perhaps using “regression models” rather than “regressions”).
Line 131–132: This sentence also needs some attention.
Line 133: I think this heading might be more confusing than enlightening. The text below is still part of the statistical methods and goes beyond individual-level covariates (e.g. software used).
Line 143: These would be characteristics of the participants not the survey itself.
Line 144: Given the repeated measures, you’ll also want to provide the number of distinct participants around here.
Line 144: This appears to be a mean and should be described as such rather than the vague term “average” (which includes means, medians, modes, and other measures of location).
Lines 174–177: This doesn’t make sense to me as currently written. Are these not results for the associations between these variables and sleep duration?
Lines 210–211: This sentence needs some attention.
Line 217: This text needs attention.
Lines 222–223: The text “sleep duration among Chinese freshman students.” suggests a reference is available for this previous work which is missing from here. Further Line 223 refers to your three previous studies but provides only two references.
Lines 231–233: Novelty, while a motivating factor, is not a strength per se. It does not make your findings more credible. You could refer to the precision of the estimates from this large sample, if you consider the 95% CI widths to be sufficiently narrow.
Lines 239–240: An effect modifier would be an interaction term. Do you mean the effect of pollution on sleep might be different for those who study more or less here? Or do you mean that study time might be a confounder if pollution is seasonally associated with examinations in the academic year?
Line 247: “affects” rather than “effect”. Also, did you not adjust for temperature in the models? This would, subject to residual confounding, remove it as a potential explanatory factor for the association.
Line 256: You refer to “month” here, but the CPSQI is for the previous week is it not?
Table 1: I appreciate it’s not difficult to guess, but please indicate either which p-values come from t-tests and Chi-squared tests or add a note that the former was used for continuous variables and the latter for categorical variables. Also, some of these measures are at the person-level (e.g. sex and I would include first measured age, BMI, smoking status, etc.) and some are at the survey cohort-level (e.g. sleep, but you could also include other non-constant measure here) and I think it might be easier to divide the table into these two groups. The first table, looking at participants would show the person-level measures for all distinct people; the second table, would look at survey cohort -specific measures (e.g. sleep). At the moment, the percentage of men is, I think, the percentage of measurements from men and not the actual percentage of men in the sample. Another option would be to clearly divide the table into two with baseline sex, age, etc. in the top half and the survey cohort sleep means in the bottom half. Whatever approach you take, it should at least be clear how many distinct people are involved and what percentage are men and women.
Table 2: “0.00 0.00” needs formatting (bottom left-hand corner).
Table 3: Refers to “# Observations (# participants)” but only lists one number for each model, which seems to be observations. Again, adding a column for the sex interaction for each predictor would strengthen the findings considerably for little extra effort with the statistical models. Rather than “Sleeping in last week (hr/day)”, I think “Sleep in last week (hr/day)” would be a more natural expression.
Author Response
Our Responses to “Reviewer(s)' Comments to Author:”
Reviewer #2: General Comments:
Point 1. Thank you for your constructive responses to my comments. The manuscript reads much better now. There are still quite a few language issues and I’ll note some of these below among my specific comments. I will inevitable have missed some of these issues in those comments though and careful proofreading will still be required to identify them.
Response: We thank the reviewer for his/her positive feedback. We have taken the comments/suggestions very seriously and have made all corresponding modifications.
Point 2. First, though, unfortunately, the modelling used here is still not at all clear to me. The only sentence about the statistical methods in the abstract is Line 25: “Linear individual fixed-effect regressions were used to analyze the data.” which would suggest that analyses were conducted at the individual level and were fixed effects only, which wouldn’t make sense. Line 163 repeats this reference to “linear individual fixed-effect regressions”.
Response:We thank the reviewer for rising this important concern. We have revised the sentence “Linear individual fixed-effect regressions” as “ Multilevel mixed-effects linear regression models” in the abstract and in the result . See Line 25 and Line 174.
Point 3. Back in the statistical methods, Lines 113–114 refers to RM-ANOVA but I cannot see where these were used (or think why they would be needed). (Note also that the t-tests on Line 114 make assumptions about normality and, in its default form, homogeneity of variance for residuals and these diagnostics need to be described here to reassure the reader. Based on data in Table 1, some of the models will not pass diagnostic checks and alternative approaches will be needed such as Mann-Whitney U.)
Response: We thank the reviewer for her/his helpful comment. We agree with the reviewer that using Mann-Whitney U in a similar nonparametric test for two independent samples. We analyzed the data again using the Mann–Whitney U test. See Line 114-116, Table 1 and online supplement.
Point 4. Lines 114–115 and 124–126 do not make it explicit what the random effects are (even if they imply what they might be). Given this description and the data, I don’t think it would be obvious to a (bio)statistician what you have done here and further explanation is definitely needed. My suggestion would be to include sufficient information so that if I, or another (bio)statistician, had your data and the statistical methods text, it would be possible to replicate your results without needing to make guesses or take a trial and error approach. The best way to do this, in my opinion, would be to include Stata code as an online supplement as this is unambiguous and complete.
Response: We thank reviewer for the comment. We have uploaded our Stata code (dofile for Table 1-2 and Table 4) as an online supplement as suggested. See Line 125-131 and Online supplement.
Point 5. The final sentence here (Lines 126–127) could be deleted or reworded to note that such an approach is required (not simply useful) in the presence of hierarchical or clustered data.
Response: We thank reviewer for the comment. We have deleted the sentence as suggested. See Line 127-131.
Point 6. In the discussion, on Lines 257–259, you restate some of the points I made last time but this leaves me confused as to what you actually did. I suspect that a very clear statement of the models, including random effects (where these are explicitly described) and/or Stata code will help me and other readers to understand the models used.
Response: We thank Reviewer for his/her helpful suggestion. We have deleted the sentence in the discussion section. We have uploaded our Stata code for using the multilevel mixed-effects linear regression models in table 3. See Line 264 and Online supplement Stata code.
Point 7. Related to this, please also indicate any specific options chosen, including the random effects estimation approach (e.g. REML). As with the t-tests, model diagnostics for the mixed effects models need to be described for the reader’s reassurance, here including normality (for residuals and BLUPs), homoscedasticity, and linearity. The last of these could be particularly important as the association could be rapid growth to an asymptote, sigmoidal, or exponential. Finally, it is traditional to note the level of statistical significance used along with a statement for how and why multiplicity was addressed or not addressed when looking at multiple models.
Response: Thanks for the comment. We have added the model details as suggested. See Line 127-131.
Point 8. Just as a suggestion for you to consider, I wonder if you could add a figure showing the means for each exposure and the means for sleep duration for each cohort (basically a pictorial representation of the overall column from the bottom of Table 1 and some rows from Table 2). This might help to give readers some idea whether the findings are likely to be from ecological confounding or reflect an actual exposure-outcome relationship. This could even allow you to replace some of the tables with figures. I’ll leave this as a suggestion for you to decide on, perhaps with input from the editor.
Response: Thanks for the suggestion. We have added figure 2 in the manuscript. See Figure 2 in Line 187.
Point 9. Note that Lines 172–173: “The estimated associations of air pollution on individual-level sleep outcomes proved there to be a more significant reduction in sleep duration among females when compared to males.” would require interaction p-values to support unless it was qualified to the much weaker claim about effect sizes being numerically (and without claiming or suggesting that this is statistically significantly) greater in one compared to the other.
Response: Thanks for the comment. We have deleted the sentence in the result section. See Line 182.
Point 10 to Point 40 and Our Responses/Revisions
We thank Reviewer 2 for his/her specific comments and suggestions. The changes have been made as suggested throughout the manuscript. Because Reviewer 2’s comments from Point 10 to Point 40 are all related issues with the manuscript’s specific points, sentence structure and grammar errors, to make it easy to follow, we provided a table below to show our point by point responses/revisions to these comments:
Point # |
Review 2’s Comments |
Our Responses/Revisions |
Point 10 |
Did you consider looking at all pollution indices in the same model. Lines 194–196 invite the reader to wonder if there is one or several pollution indices that explain the association with correlations between the indices leading to the pattern of statistically significant results. |
Thanks for the comment. We have revised the sentence. See Line 205-206. |
Point 11 |
Line 23: “...was gathered…” should be “…were gathered…” (plural to match “levels” on Line 22). |
The change has been made as suggested. See Line 23. |
Point 12 |
Lines 27–28: As you are talking about decreases in absolute terms, the CI limits should go from smaller in absolute value to larger in absolute value (i.e. “…by 0.68 (95% confidence interval [CI] = 0.63, 0.73)”. Related to this, on Line 166, the CI limits for the decrease don’t need negative signs as the direction is also made clear by “reduction” on Line 165. See also Lines 169 and 171. |
Thanks for the comment. The change has been made as suggested. See Line 27-28, Line 177, Line 180 and 182. |
Point 13 |
Line 36: “estimates” would be better here than “approximates”. |
The change has been made as suggested. See Line 37. |
Point 14 |
Line 42: The exponent “3” isn’t superscript (as it is on Line 38). |
The change has been made as suggested. See Line 43. |
Point 15 |
Lines 50–51: I’m not sure how it can be both “a significant health concern” and “a public health epidemic”. An epidemic in this non-communicable sense is more serious than a significant health concern to me. |
Thanks for the comments. We have revised the sentence and deleted “a public health epidemic”. See Line 51-52. |
Point 16 |
Line 88: “test-restes reliability” should be “test-retest reliability”. |
The change has been made as suggested. See Line 89. |
Point 17 |
Line 88: You need to be specific about what the validity of 0.64 represents (internal consistency through alpha, correlation with a gold-standard measure, etc.) |
Thanks for the comments. We have added validity measurement. See Line 89-90. |
Point 18 |
Lines 107–109: But keep in mind that these are highly positively skewed measurements, as indicated by the SD relative to the mean, and so don’t capture “typical” very well, although they do capture cumulative exposure. You could consider, if you wanted, adding median values to these to indicate “typical” potential exposures. The same issue arises in Table 2. |
Thanks for the suggestion. We used “mean” instead of “median” based on the following considerations: The data on average air pollution concentration were standardized through demeaning (i.e., subtracting the mean from each value) and then dividing by its own standard deviation (i.e., PM2.5 z-scores) Therefore, no change was made. |
Point 19 |
Line 112: This isn’t a sentence (“Means, SD, and percentages summarized and compared for characteristics of the overall sample.”) |
The change has been made as suggested. See Line 114-115. |
Point 20 |
Line 119: Rather than “controlled for”, I think you might mean “were included in”. |
The change has been made as suggested. See Line 121, Line 133. |
Point 21 |
Lines 120–121: This sentence needs some attention (including for consistent hyphenation and perhaps using “regression models” rather than “regressions”). |
The change has been made as suggested. See Line 123. |
Point 22 |
Line 131–132: This sentence also needs some attention. |
The change has been made as suggested. See Line 135. |
Point 23 |
Line 133: I think this heading might be more confusing than enlightening. The text below is still part of the statistical methods and goes beyond individual-level covariates (e.g. software used) |
Thanks for the comments. We have deleted this heading. See Line 136-137. |
Point 24 |
Line 143: These would be characteristics of the participants not the survey itself. |
The change has been made as suggested. See Line 145-146. |
Point 25 |
Line 144: Given the repeated measures, you’ll also want to provide the number of distinct participants around here. |
The change has been made as suggested. See Line 78-80. |
Point 26 |
Line 144: This appears to be a mean and should be described as such rather than the vague term “average” (which includes means, medians, modes, and other measures of location). |
Thanks for the suggestion. The changes have been made as suggested. Specially, we used “mean” instead of “average”. See Line 147. |
Point 27 |
Lines 174–177: This doesn’t make sense to me as currently written. Are these not results for the associations between these variables and sleep duration? |
Thanks for the comments. We have deleted the sentence in the result section. See Line 182.
|
Point 28 |
Lines 210–211: This sentence needs some attention. |
Thanks for the comments. We have rewrited the sentence as suggested. See Line 219. |
Point 29 |
Line 217: This text needs attention. |
The change has been made as suggested. See Line 225-226. |
Point 30 |
Lines 222–223: The text “sleep duration among Chinese freshman students.” suggests a reference is available for this previous work which is missing from here. Further Line 223 refers to your three previous studies but provides only two references. |
Thanks for the comments. We have counted this study as a reference. We have revised the sentence and changed “three” to “two” previous studies. See Line 231.
|
Point 31 |
Lines 231–233: Novelty, while a motivating factor, is not a strength per se. It does not make your findings more credible. You could refer to the precision of the estimates from this large sample, if you consider the 95% CI widths to be sufficiently narrow. |
Thanks for the comments. We have revised the sentence. See Line 240. |
Point 32 |
Lines 239–240: An effect modifier would be an interaction term. Do you mean the effect of pollution on sleep might be different for those who study more or less here? Or do you mean that study time might be a confounder if pollution is seasonally associated with examinations in the academic year? |
Thanks for the comments. We have deleted the sentence. See Line 248. |
Point 33 |
Line 247: “affects” rather than “effect”. Also, did you not adjust for temperature in the models? This would, subject to residual confounding, remove it as a potential explanatory factor for the association |
Thanks for the comments. We have adjusted for temperature in the models. We have deleted the sentence. See Line 255. |
Point 34 |
Line 256: You refer to “month” here, but the CPSQI is for the previous week is it not? |
Thanks for the comments. We have revised it as refer to “week”. See Line 263. |
Point 35 |
Table 1: I appreciate it’s not difficult to guess, but please indicate either which p-values come from t-tests and Chi-squared tests or add a note that the former was used for continuous variables and the latter for categorical variables. |
Thanks for the comments.The change has been made as suggested. See Table 1 note in Line 154-155 and Table 2 note 165. |
Point 36 |
Also, some of these measures are at the person-level (e.g. sex and I would include first measured age, BMI, smoking status, etc.) and some are at the survey cohort-level (e.g. sleep, but you could also include other non-constant measure here) and I think it might be easier to divide the table into these two groups. The first table, looking at participants would show the person-level measures for all distinct people; the second table, would look at survey cohort -specific measures (e.g. sleep). At the moment, the percentage of men is, I think, the percentage of measurements from men and not the actual percentage of men in the sample. Another option would be to clearly divide the table into two with baseline sex, age, etc. in the top half and the survey cohort sleep means in the bottom half. Whatever approach you take, it should at least be clear how many distinct people are involved and what percentage are men and women. |
Thanks for the suggestion. We have separated the table into 2 table. See Table 1 and Table 2. |
Point 37 |
Table 2: “0.00 0.00” needs formatting (bottom left-hand corner). |
Thanks for the comments.The change has been made as suggested. See Table 3. |
Point 38 |
Table 3: Refers to “# Observations (# participants)” but only lists one number for each model, which seems to be observations. |
Thanks for the comments.The change has been made as suggested. See Table 3. |
Point 39 |
Again, adding a column for the sex interaction for each predictor would strengthen the findings considerably for little extra effort with the statistical models. |
Thanks for the comments. We have deleted the sex differences in the ms. Therefore, no change was made. We appreciate your suggestions. |
Point 40 |
Rather than “Sleeping in last week (hr/day)”, I think “Sleep in last week (hr/day)” would be a more natural expression. |
Thanks for the comments.The change has been made as suggested. See Table 3. |
We want to express our appreciation for Reviewer 2 so many helpful comments! Thank you very much!
Reviewer 3 Report
The authors managed to improve the manuscript significantly.
The only thing that remains confusing is the use of multiple visits. The authors used data from 5 different cohorts and included participants with at least 2 visits. Can individuals contribute to more than 1 cohort? It would be informative to add the total number of questionnaire (data rows) that have been used for the present study.
Line 89: "prsent" should be "present"
Once the design contains a bit more detail about overlap, number of visits, etc, I am oK.
Author Response
Reviewer #3: General Comments:
Point 1. The authors managed to improve the manuscript significantly.
Response: We thank Reviewer 2 for the positive feedback.
Point 2. The only thing that remains confusing is the use of multiple visits. The authors used data from 5 different cohorts and included participants with at least 2 visits. Can individuals contribute to more than 1 cohort? It would be informative to add the total number of questionnaire (data rows) that have been used for the present study.
Response: Thanks for the comment. We have added data on freshmen who completed survey only once data into the data. We also have used “multilevel mixed-effects linear regression models” instead of “Linear individual fixed-effect regressions” to analyze the data. A total of 16,889 Tsinghua University freshmen student enrolled and responded to the survey. We have added the information as suggested. See Line 19 and Line 78-80.
Point 3. Line 89: "prsent" should be "present"
Response: Thanks for the comments.The change has been made as suggested. See Line 88.
Point 4. Once the design contains a bit more detail about overlap, number of visits, etc, I am oK.
Response: Thanks for the suggestion.The change has been made as suggested. See Line 19 and Line 78-80.
Thank you very much!
Round 3
Reviewer 2 Report
Thank you for your revisions and responses. In particular, thank you very much for sharing the code that you used for the statistical analyses. This highlights an issue with your mixed models and perhaps explains some of the confusion I was having with the written descriptions of these models. You say (Lines 126–128) “There were also two levels of non-independence (person-level and cohort week-level) in the model. We chosen the random effects estimation methods using maximum likelihood (ML).”, which is a valid approach (but note that “chosen” here should be “chose” and “methods” should be “method”) but the code, e.g. “xtmixed $ylist $xlist ||surveyyear:” incorporates only a single level of non-independence through the random effect for survey. At the moment, multiple surveys of the same person are treated as if they were surveys of different people, violating the assumption of no unmodelled dependence in the data. If these statistical models are not those that you are familiar with, you might like to talk to a local (bio)statistician to get some advice around how to address these crossed random effects (person and survey). The statistical methods also still need to discuss the model diagnostics used (for the t-tests and the mixed models)—see Points 3 and 7 from last time.
For Table 1, there is a note “a p-values come from t-tests or the Mann–Whitney U test.” but this will need two separate symbols to distinguish between these. A similar point applies to Table 2.
In Table 1, you give the n=31806 observations but you have 16889 (Line 78) participants. Counting people multiple times doesn’t make sense here (see Point 36 from last time), particularly when you are comparing them (t-tests, Mann-Whitney U, and Chi-squared tests all assume independence of observations, which would not hold here).
Related to the above, Table 2 could show the number of observations for each survey-sex combination, which would add to n=31806.
Table 3 (air pollution) could then show the number of people for each survey (both sexes), which again would add to n=31806.
Table 3 (regression results, should be numbered Table 4) still needs to indicate the number of people in each analysis (see Point 38 last time—my apologies if it wasn’t clear but I was hoping you would add the number of people here). I appreciate that this is stated in the text, but the number of people is at least, if not more, important as the number of observations and so both should be shown in the table. If these numbers are the same for all four dependent variables, which seems to be the case, these sets of numbers might work better as a note underneath Table 4.
Author Response
Our Responses to “Reviewer(s)' Comments to Author:”
Reviewer #2: General Comments:
Point 1. Thank you for your revisions and responses. In particular, thank you very much for sharing the code that you used for the statistical analyses.
Response: We thank the reviewer for his/her positive feedback. We have taken the comments/suggestions very seriously and have made all corresponding modifications.
Point 2. This highlights an issue with your mixed models and perhaps explains some of the confusion I was having with the written descriptions of these models. You say (Lines 126–128) “There were also two levels of non-independence (person-level and cohort week-level) in the model. We chosen the random effects estimation methods using maximum likelihood (ML).”, which is a valid approach (but note that “chosen” here should be “chose” and “methods” should be “method”) but the code, e.g. “xtmixed $ylist $xlist ||surveyyear:” incorporates only a single level of non-independence through the random effect for survey. At the moment, multiple surveys of the same person are treated as if they were surveys of different people, violating the assumption of no unmodelled dependence in the data. If these statistical models are not those that you are familiar with, you might like to talk to a local (bio)statistician to get some advice around how to address these crossed random effects (person and survey).
Response: We thank the reviewer for her/his helpful comment. We have revised the English editing “chose” and “method” as suggested. See Line 127-128. We also thank the reviewer for the code concern. We used the code “xtmixed $ylist $xlist ||surveyyear:” is the same as the code “xtmixed $ylist $xlist ||surveyyear:||id:”. We apologized that make you some of the confusion. The code “surveyyear” indicated cohort week and the code “xtmixed” included person id. Therefore, there were two levels of non-independence (person-level and cohort week-level) in the model. Again, we apologized for your confusions in the code.
Point 3. The statistical methods also still need to discuss the model diagnostics used (for the t-tests and the mixed models)—see Points 3 and 7 from last time.
Response: We have added the model diagnostics used as suggested. See Line 131-132.
Point 4. For Table 1, there is a note “a p-values come from t-tests or the Mann–Whitney U test.” but this will need two separate symbols to distinguish between these. A similar point applies to Table 2.
Response: We thank reviewer for the comment. We have revised it as suggested. See table 1 and Table 2.
Point 5. In Table 1, you give the n=31806 observations but you have 16889 (Line 78) participants. Counting people multiple times doesn’t make sense here (see Point 36 from last time), particularly when you are comparing them (t-tests, Mann-Whitney U, and Chi-squared tests all assume independence of observations, which would not hold here).Related to the above, Table 2 could show the number of observations for each survey-sex combination, which would add to n=31806.
Response: We thank reviewer for the comment. We have added the number of oberservations for each survey. See Table 2.
Point 6. Table 3 (air pollution) could then show the number of people for each survey (both sexes), which again would add to n=31806.
Response: We thank Reviewer for his/her helpful suggestion. We have showed the number of people for each survey as suggested. See table 3.
Point 7. Table 3 (regression results, should be numbered Table 4) still needs to indicate the number of people in each analysis (see Point 38 last time—my apologies if it wasn’t clear but I was hoping you would add the number of people here). I appreciate that this is stated in the text, but the number of people is at least, if not more, important as the number of observations and so both should be shown in the table. If these numbers are the same for all four dependent variables, which seems to be the case, these sets of numbers might work better as a note underneath Table 4.
Response: We thank reviewer for the comment. The change has been made as suggested. See Table 4. We have added a note underneath Table 4. “There were 16,870 participants for total, 11,452 participant for male and 5,538 participants for female in the regression.”
We want to express our appreciation for Reviewer 2 so many helpful comments from the bottom of my heart!

This manuscript is a resubmission of an earlier submission. The following is a list of the peer review reports and author responses from that submission.
Round 1
Reviewer 1 Report
Thank you for interesting manuscript! However, there are couple of issues which need to be solved before accepted for publications. In following I enlist the major ones:
in Introduction you mention "..few studies..." but after either provide no reference or one reference only! That is not "few"! It is none or one...please add references if they exists
in Table 1, do the numbers for smoking and drinking mean that it is only 73 men who smoke and 365 who drink?! That sounds excellent, but is it really the truth?
you do not specify whether the air population is ambient air or indoor air? I think in case of sleep disturbance it is important to make this distinction
though the air pollution data are form the meteorological service, it would be important to know more about the method of measurement and especially the location of measurement point with respect to places where sleep disturbance was assessed
I am not fully convinced with your statistical analysis; could you please describe the linear regression model more in depth? How do you get from coefficients around -0.66 to time of about 0.25 hours less sleep?
because of student study population, did you collect data upon daily study time? I mean especially self-learning practices at evenings and nights...Those might be very important effect modifiers
Author Response
Responses to reviewers' comments:
Reviewer #1:
Reviewer’s comment 1:
Thank you for interesting manuscript! However, there are couple of issues which need to be solved before accepted for publications. In following I enlist the major ones:
Response: We are glad that that reviewer 1 found this manuscript interesting. We have revised the manuscript as suggested, and we believe this manuscript has improved. We also want to thank Reviewer 1 for providing the helpful comments/suggestions for the manuscript.
Reviewer’s comment 2:
in Introduction you mention "..few studies..." but after either provide no reference or one reference only! That is not "few"! It is none or one...please add references if they exists.
Response: We thank the reviewer for the comment. We have taken the comments/suggestions very seriously and have re-checked the reference. To our best knowledge, this is the first study to examine the relationship between ambient air pollution and sleep duration. We have revised it as suggested. See Page 2, Line 60-62.
Reviewer’s comment: 3. in Table 1, do the numbers for smoking and drinking mean that it is only 73 men who smoke and 365 who drink?! That sounds excellent, but is it really the truth?
Response: Thank you to the reviewer for the comment. We have checked the self-reported smoking and drinking number, and the number is right. The reason why the numbers for smoking and drinking are so low is because all participants were freshmen at the Tsinghua University, the #1 ranked university in China. Another possible explanation for this low number is that all freshmen self-reported their smoking and drinking habits and thus is subject to social desirability bias.
Reviewer’s comment: 4. you do not specify whether the air population is ambient air or indoor air? I think in case of sleep disturbance it is important to make this distinction.
Response: We thank the reviewer for the comments. We used the outdoor ambient air pollution data. We agree with the reviewer’s comments that it is important to make the distinction between ambient (indoor) and ambient (outdoor) air pollution and is important when referencing sleep disturbance. We added the limitation in this part. See Page 9, 255-257.
Reviewer’s comment: 5. though the air pollution data are form the meteorological service, it would be important to know more about the method of measurement and especially the location of measurement point with respect to places where sleep disturbance was assessed
Response: We thank the reviewer for the comment. We agree with the comment. In the limitation part, we stated the limitation. “Beijing, a metropolitan area, likely has more than one level of air pollution concentration, and thus requires a method to average air pollution levels over multiple areas. We computed daily mean air pollution concentrations using the city-average air pollution measure. This process may have covered up multiple local variations and likely masked the individual variations in the uptake of pollution”.
Reviewer’s comment: 6. I am not fully convinced with your statistical analysis; could you please describe the linear regression model more in depth?
Response: We thank Reviewer 1 for the comment. In statistics, a fixed effects model is a statistical model in which the model parameters are fixed quantities. In panel data where longitudinal observations exist for the same subject, fixed effects represent the subject-specific means. In panel data analysis, the term fixed effects estimator (also known as the within estimator) is used to refer to an estimator for the coefficients in the regression model including those fixed effects (one time-invariant intercept for each subject). In our statistical analysis, individual fixed-effect regression was selected because it only used within-individual variations in total daily hour of sleep to identify the impacts of air pollution concentration, thus removing potential omitted variable bias due to differences in time-invariant individual characteristics such as genes, gender, ethnicity, habits, and personal preferences. Due to the exclusive dependence upon within-individual variations in an outcome measure, individual fixed-effect regressions could only estimate the effect of a time-variant independent variable. Therefore, time-invariant individual characteristics such as gender and ethnicity were not examined.
Reviewer’s comment: 7. How do you get from coefficients around -0.66 to time of about 0.25 hours less sleep?
Response: Thank you very much for the reviewer’s comments. Our results suggested air pollution in PM2.5 increased by 78.85 µg/m³ and led to a decrease in sleep duration by 0.66 hour (39.6 minutes) per day among freshmen. Our results are consistent with previous research by Fang et al. showing that traffic-related air pollution increase in annual black carbon by 0.21µg/m³ were associated with 0.23–0.25 hours per day less sleep American adults aged 53.8 years on average.
Reviewer’s comment: 8. because of student study population, did you collect data upon daily study time? I mean especially self-learning practices at evenings and nights...Those might be very important effect modifiers.
Response: We thank the reviewer for the comment. The change has been made as suggested. We have added the limitation. See Page 9, Line 245-247. “In addition, because of the student study population, we did not collect data upon daily study time. Self-learning practices at nights may be very important effect modifiers”.
Reviewer 2 Report
A one-time/year paper-pencil based health survey of all freshmen at the Tsinghua University from 2013-2018 to evaluate the participants’ physical and mental health condition was evaluated against environmental air quality measurements from a Ministry of Environmental Protection site. A longitutinal analysis using linear individual fixed-effect regressions to examined the effects of air pollution concentration on individual-level outcomes in sleep. The methodology is well described and appropriate. As the data was not collected for this purpose and the air quality data is from a fixed site - not specific to any individual, the limitations are discussed.
There are no methodological comments. However, the paper needs moderate editing for english language grammar and structure which are not done as part of this review. A few other minor comments follow:
- line 40 - indicate air pollutant type - ie. PM10.
- line 54 - ...more than 34% adult were less than 7 h of sleep in..., think author means 'more than 34% adult had less than 7 h of sleep per day/on average?
- line 204 - Repeat from above paragraph line 198.
- line 233-234 - Not sure I understand what is trying to be said?
Author Response
Reviewer #2:
Reviewer’s comment: 1. A one-time/year paper-pencil based health survey of all freshmen at the Tsinghua University from 2013-2018 to evaluate the participants’ physical and mental health condition was evaluated against environmental air quality measurements from a Ministry of Environmental Protection site. A longitutinal analysis using linear individual fixed-effect regressions to examined the effects of air pollution concentration on individual-level outcomes in sleep. The methodology is well described and appropriate. As the data was not collected for this purpose and the air quality data is from a fixed site - not specific to any individual, the limitations are discussed.
Response:
We are glad that that reviewer 2 found this manuscript’s methodology well described and appropriate. We also want to thank Reviewer 2 for providing the detailed and helpful comments/suggestions for the manuscript.
Reviewer’s comment: 2.There are no methodological comments. However, the paper needs moderate editing for english language grammar and structure which are not done as part of this review.
Response: Thanks for the comments. A native English speaker who is our coauthor Shelby Paige Gordon again has proofread the revised manuscript. The sections of the manuscript that were edited are colored in yellow. We believe the readability of this manuscript has improved as a result.
Reviewer’s comment: 3. - line 40 - indicate air pollutant type - ie. PM10.
Response: We thank Reviewer 2 for the comment. We have revised the manuscript as suggested. See Page 2, Line 38.
-Reviewer’s comment: 4. -line 54 - ...more than 34% adult were less than 7 h of sleep in..., think author means 'more than 34% adult had less than 7 h of sleep per day/on average?
Response: We thank Reviewer 2 for the comment. We have revised the manuscript as suggested. See Page 2, Line 52-53.
Reviewer’s comment: 5. - line 204 - Repeat from above paragraph line 198.
We thank Reviewer 2 for the comment. We have deleted the repeated paragraph in line 189.
Reviewer’s comment: 5. - line 233-234 - Not sure I understand what is trying to be said?
Response: Thank you very much for the reviewer’s comments. We have revised the paragraph in line 221-222.
Again, we want to express our gratitude to both Reviewer 1 and Reviewer 2’s meaningful comments!
Round 2
Reviewer 1 Report
I still have some issues with the Method and Result part.
Now you wrote in your comments, that air pollution (exposure) data are calculated not only as a mean value over days of the week, but also across Beijing and many measurement stations. We see huge variations in air pollution over days, but in your Method in the manuscript you do not even mention calculating the mean also across Beijing! How many places in Beijing? How did you get the mean over them? What is the variation of values of air pollution across places? Is using the mean a correct approach in this case? Shouldn't the closest measurement station data be more relevant instead of mean across Beijing? Is the mean value you are using in your regression valid under these cuircumstances?
Related to this and to your regression model, is it correct to say, that by using the mean of an air pollution indicator you have in principle one value for the indicator and about 3000 values of sleep duration for each cohort? Can you elaborate a bit on this how do you do the regression? Do you merge the five cohorts into one regression and by this have five exposure measurement points?
The second point is still the issue of getting from regression coefficient to hours of shorter sleep; you answer by giving a reference to other study, but that was not my point. You need to explain how do you transfer the coefficient into minutes.
Author Response
Reviewer’s comment 1:
Now you wrote in your comments, that air pollution (exposure) data are calculated not only as a mean value over days of the week, but also across Beijing and many measurement stations. We see huge variations in air pollution over days, but in your Method in the manuscript you do not even mention calculating the mean also across Beijing! How many places in Beijing? How did you get the mean over them? What is the variation of values of air pollution across places? Is using the mean a correct approach in this case? Shouldn't the closest measurement station data be more relevant instead of mean across Beijing?
Response: We thank the reviewer for the comments. We want to express thankful and agree to the reviewer’s concern. About the air pollution measurment, we used air pollution data of the mean across Beijing by the Ministry of Environmental Protection of the People’s Republic of China. We can not collected the air pollution data from the clostest measurement station data in our study. It is the limitation of our study, we stated the point in the limitation part. See Page 9 Line 258-264.
“Collecting daily mean air pollution concentrations using the city-average air pollution measure is another limitation. This process may have covered up multiple local variations, and was likely masked the individual variations in the uptake of pollution. Beijing , a metropolitan area, likely has more than one levels of air pollution concentration and many air pollution measurement stations, and thus requires collecting the closest measurement station data be more relevant instead of mean across Beijing.”
Reviewer’s comment: 2. Is the mean value you are using in your regression valid under these cuircumstances? Related to this and to your regression model, is it correct to say, that by using the mean of an air pollution indicator you have in principle one value for the indicator and about 3000 values of sleep duration for each cohort? Can you elaborate a bit on this how do you do the regression? Do you merge the five cohorts into one regression and by this have five exposure measurement points?
Response: Based on ecological study design, we perfomed all statatistical regression in Stata 14.2 SE version (StataCorp, College Station, TX). We used linear individual fixed-effect regressions used z-score (standardized through demeaning, i.e., subtracting the mean from each value) and the repeated-measure survey sleep data. We merge the five cohorts into one regression. Our stata dofile example in the stata is in the following.
* Add weekly average pm2.5
gen pm25 =.
replace pm25 = 28.75 if order == 201301
replace pm25 = 52.80 if order == 201304
replace pm25 = 178.71 if order == 201401
replace pm25 = 69.59 if order == 201402
replace pm25 = 36.06 if order == 201403
replace pm25 = 89.24 if order ==201501
replace pm25 = 43.43 if order ==201502
replace pm25 = 84.43 if order ==201601
replace pm25 = 57.71 if order ==201602
replace pm25 = 37.43 if order ==201701
replace pm25 = 34.29 if order ==201702
* Add weekly average pm2.5SD
gen pm25sd =.
replace pm25sd = (28.75-73.61)/78.85 if order == 201301
replace pm25sd = (52.80-73.61)/78.85 if order == 201304
replace pm25sd = (178.71-73.61)/78.85 if order == 201401
replace pm25sd = (69.59-73.61)/78.85 if order == 201402
replace pm25sd = (36.06-73.61)/78.85 if order == 201403
replace pm25sd = (89.24-73.61)/78.85 if order == 201501
replace pm25sd = (43.43-73.61)/78.85 if order == 201502
replace pm25sd = (84.43-73.61)/78.85 if order == 201601
replace pm25sd = (57.71-73.61)/78.85 if order == 201602
replace pm25sd = (37.43-73.61)/78.85 if order == 201701
replace pm25sd = (34.29-73.61)/78.85 if order == 201702
We performed Fixed-effect model, the Stata command to run fixed effecst is xtreg.
Fixed-effects (FE) model
xtreg depvar [indepvars] [if] [in] [weight] , fe [FE_options]
*PM25 and Sleeping behavior
sort $id $time
xtset $id $time
* Fixed effects or within estimator
xtreg $ylist $xlist, fe
Reviewer’s comment: 3. The second point is still the issue of getting from regression coefficient to hours of shorter sleep; you answer by giving a reference to other study, but that was not my point. You need to explain how do you transfer the coefficient into minutes.
Response: We thank for reviewer’s comments. The regression coefficient is hour, -067 hour is equal to -40.2 minutes (-0.67 hour*60 mintues/hour =-40.2 minutes) , that is an increase in air pollution concentration in PM2.5 by one SD (i.e., PM2.5 by 78.85 µg/m³) was associated with a reduction in daily hours of sleep by 40.2 minutes.
Again, thank you very much!
Round 3
Reviewer 1 Report
I am sorry, I think the air pollution measurement as presented, is so in-precise due to presenting one mean value for a large city like Beijing, moreover without even knowing how many stations contribute to that value and what is the distribution of values across the city, that it is more then a limitation. It makes all your estimates very questionable!
Other point is the Air Quality index; an index is usually a number on a certain scale. You present it in micrograms/m3. It would be necessary to explain this.
I would also like to see the regression formula, not a Stata output, but the formula because to my knowledge a regression coefficient is a coefficient and not time.
I think the methodology needs to be improved substantially before the manuscript would be accepted.